# Rethinking Deep Thinking: Stable Learning of Algorithms using Lipschitz Constraints

**Jay Bear**      **Adam Prügel-Bennett**      **Jonathon Hare**
The University of Southampton, Southampton, UK
{jdh1g19,abp1,jsh2}@soton.ac.uk

## Abstract

Iterative algorithms solve problems by taking steps until a solution is reached. Models in the form of Deep Thinking (DT) networks have been demonstrated to learn iterative algorithms in a way that can scale to different sized problems at inference time using recurrent computation and convolutions. However, they are often unstable during training, and have no guarantees of convergence/termination at the solution. This paper addresses the problem of instability by analyzing the growth in intermediate representations, allowing us to build models (referred to as Deep Thinking with Lipschitz Constraints (DT-L)) with many fewer parameters and providing more reliable solutions. Additionally our DT-L formulation provides guarantees of convergence of the learned iterative procedure to a unique solution at inference time. We demonstrate DT-L is capable of robustly learning algorithms which extrapolate to harder problems than in the training set. We benchmark on the traveling salesperson problem to evaluate the capabilities of the modified system in an NP-hard problem where DT fails to learn.

## 1   Introduction

Iteration is a key ingredient in a vast number of important algorithms. Incremental progress towards a solution is demonstrated in many of these. Well-known examples include insertion sort, gradient descent, and the simplex method. This paper explores models that learn iterative algorithms. We ask questions including: Can deep learning models be used to learn the complex steps of algorithms similar to these? Is there a way to guarantee a solution or approximation is reached? Can we learn algorithms that extrapolate to larger or harder instances than we train on? In addressing these questions we propose a model called Deep Thinking with Lipschitz Constraints (DT-L).

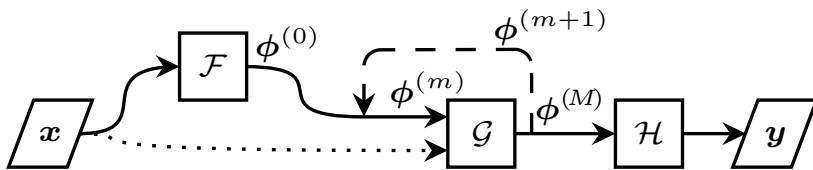

Figure 1: Recurrent-based model architectures for learning algorithms with input $x$ and output $y$. $\mathcal{F}$, $\mathcal{G}$ and $\mathcal{H}$ are convolutional networks that work on any size input. A scratchpad $\phi$ serves as the working memory during computation. As described in Section 2 the original DT model didn't include *recall*, denoted by the dotted line. The improved DT-R and our DT-L model include this connection.

Some work towards answering the above questions has occurred in recent years. A key approach has been to utilize various forms of a learned recurrent function to implement a 'step' in an iterative algorithm that computes a solution to a problem. Typically these recurrent functions are combined

38th Conference on Neural Information Processing Systems (NeurIPS 2024).

with learned functions that pre-process the initial input, and post-process the output of the last iteration into the final solution form. Examples of such approaches are the Deep Thinking (DT) networks [18] and Deep Thinking with Recall (DT-R) networks [2] that we build upon in this work (see Figure 1).

There are two key features to these models: firstly the use of an iteratively updated *scratchpad*, $\phi$ serving as a working memory; and secondly, the construction of the model from convolutional layers to enable it to scale to arbitrary sized problems. The latter is important because in principle it allows training on smaller problems, and then extrapolation at inference time to larger problems [2, 18].

Many challenges still remain with the existing DT and DT-R approaches however, and we aim to address these in this work. Firstly, DT-style networks are quite difficult to train and can be very unstable both during training and inference. The DT-style networks previously demonstrated in the literature are massively overparameterized — we show in Section 3 that model width is inversely related to training stability in existing models, and later demonstrate how this can be addressed (see Section 4). Secondly, existing approaches have no guarantees on the convergence of the learned algorithm; this is important because if a solution to a problem exists, one would hope that an algorithm to solve it would terminate or reach a stable state once the solution has been reached. Section 4 details a design approach to guarantee convergence within the limits of floating point precision by considering the network in terms of a discrete dynamical system. Our contributions are:

- We systematically analyze DT networks and explore why they can be unstable;
- we introduce techniques grounded in theory into the DT framework which improve the stability of learning and guarantee convergence at run time. This allows the use of much smaller models to tackle the same problems, and improves extrapolation performance;
- we propose our Deep Thinking with Lipschitz Constraints (DT-L) model and perform a comprehensive evaluation and ablation study; and,
- we show that the approach can be applied to learn to find low-cost tours for a range of TSPs, including non-Euclidean and asymmetric instances.

## 2    Related work

Deep Thinking (DT) networks [18] were designed to learn algorithms by using recurrence to induce iterative behavior in such a way they could be trained on smaller and simpler examples before extrapolating to larger tests. The networks were used to solve 'Easy-to-Hard' problems (consisting of prefix sums, mazes, and chess puzzles [17]) and they could be trained with few iterations on easy problems in such a way that they can solve harder problems by increasing the number of iterations.

DT networks were further improved by Bansal et al. [2] by means of adding 'recall' (which we refer to as DT-R networks) — a mechanism for the recurrent component to have continuous access to the original input. Recall successfully allowed DT-R networks to solve much larger tests than DT networks without recall, while also mitigating *overthinking*, where if after too many iterations at inference time the predicted solution becomes progressively worse. In addition to recall, Bansal et al. introduced *incremental progress training* (IPT), a training method which disables gradient tracking for a random number of initial iterations. This prevents the model from learning behaviors based strictly on the number of iterations and instead promotes incremental modification to the internal states.

There exist alternative approaches to producing machines which can learn algorithms. One set of models are Neural Turing Machines (NTMs), which use attention in a recurrent system for reading to, and writing from, some external memory [6]. NTMs showed success in learning cell-based copying and repeating tasks which could be applied to unseen inputs. Differentiable Neural Computers (DNCs) operate similarly to NTMs, differing primarily in their method of external memory access [7]. In the NTM, like most RNNs, the amount of compute is directly tied to the input sequence length. Graves [5] proposed a method to adaptively select the compute budget in RNNs. The models studied here are different in the sense that the objective is to extrapolate beyond the training data to harder problem instances as the compute budget is increased, and the input and output is not a sequence.

A second class of models exist where a specific base iterative algorithm is defined, and the parameters of the algorithm, or the problem, are learned. Examples include the reverse diffusion process learned in diffusion models [9], and models which involve optimization using gradient descent iterations

within the forward pass to generate an output with certain properties that are difficult to produce directly with a neural network, such as permutation equivariance [21]. Our work does not constrain the class of algorithm used directly, other than enforcing that we learn an iterative one.

# 3 Analysis of Deep Thinking Networks

The main architecture used in pre-existing Deep Thinking networks [2, 18], and our own network, is shown in Figure 1. It takes an input problem instance $x$ and generates a solution $y$. The model consists of three convolutional networks $\mathcal{F}$, $\mathcal{G}$ and $\mathcal{H}$. The function $\mathcal{F}$ is responsible for initially pre-processing the input $x$ into the initial state $\phi^{(0)}$; function $\mathcal{G}$ is the recurrent function that takes the current state, $\phi^{(m)}$ (plus the original input $x$ in the case of DT-R and our models), to produce the next state, $\phi^{(m+1)}$. The final state produced by $\mathcal{G}$, after $M$ iterations, is denoted $\phi^{(M)}$. The function $\mathcal{H}$ takes $\phi^{(M)}$ and produces the predicted output $y$. Formally

$$\phi^{(0)} = \mathcal{F}(x) \,, \tag{1}$$

$$\phi^{(m+1)} = \mathcal{G}\big(\phi^{(m)}, x\big) \quad \forall m \in \{0, \ldots, M-1\} \,, \tag{2}$$

$$y = \mathcal{H}\big(\phi^{(M)}\big) \,. \tag{3}$$

Architecturally these networks are recurrent neural networks. Unlike more commonly used recurrent networks such as LSTMs they are not used to tackle problems with sequential data, but rather the recurrent part is used to find a solution through an iterative process. The *algorithm* that the recurrent network uses, as well as the pre- and post- processing functions are learned through supervised training on example input-solution pairs for a particular problem.

As the networks $\mathcal{F}$, $\mathcal{G}$ and $\mathcal{H}$ are convolutional neural networks they can work with an arbitrary size input. The solution $y$ is typically the same size as the input $x$. The feature of DT and DT-R that excited interest was that, not only could they solve unseen problem instances of the same size as they were trained on (which we call *interpolation*), but when trained on small problem instances (with relatively small $M$), they were able to find low cost solutions on much larger problem instances (which we refer to as *extrapolation*) by increasing $M$ at inference time. Both DT and DT-R suffer from poor stability both in training but particularly at inference time when extrapolating to larger problems. This often lead to overflow errors. This was particularly seen if the number of channels in network $\mathcal{G}$ is reduced, then DT and DT-R are hard to train with many runs failing to find a solution. We show the reason for this is that there is no mechanism to control the change in size of the scratchpad representation, $\|\phi^{(m)}\|/\|\phi^{(m-1)}\|$, leading to this either overflowing or vanishing during learning.

## 3.1 Training Stability

In this section we focus on DT-R as this is more stable than the original DT network. We use the *spectral norm* of the reshaped weights of a convolution[1] to capture the expansion or shrinkage in magnitude of the output relative to the input of a sub-network, with a value of 1 meaning that the

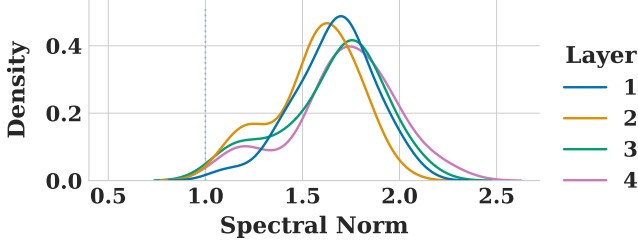

Figure 2: Distribution of spectral norms of reshaped weight matrices for the different convolutional layers in the recurrent part of DT-R. 30 prefix-sum-solving models with width $w = 32$ were sampled.

---

[1]Given a convolutional layer with weights shaped $(C_{out}, C_{in}, n)$, where $n$ is the number of elements (e.g. the kernel height times width in a 2D convolution) we can flatten the weights into a matrix of shape $C_{out} \times C_{in} \cdot n$; the spectral norm is the largest singular value of this matrix.

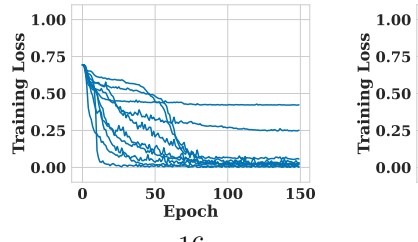 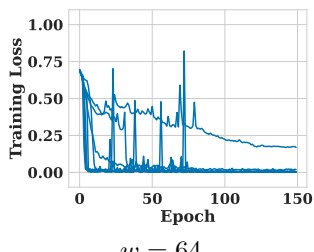 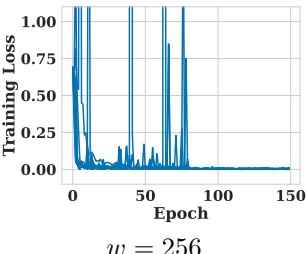

$$w = 16 \qquad\qquad w = 64 \qquad\qquad w = 256$$

Figure 3: Mean training (cross-entropy) loss at each epoch for prefix-sums-solving models of varying width $w$. For small $w$ training is stable, but not all models converge; larger $w$ has a higher chance of models reaching a small loss, but the training process has very large spikes in the loss which causes some models to explode. Each curve is measured from a different random initialization of the model throughout training, for 10 models of each width

magnitude is constant. In Figure 2, we use violin plots to show the distribution of spectral norms for the four convolutional layers that arise after training DT-R on the prefix-sum problem (see Section 5 for a description of the problem). We observe that the spectral norms are typically greater than one, which can lead to the norm of $\phi^{(m)}$ growing with each iteration. In turn this can cause the model to overflow when applying the model to a larger input (the extrapolation scenario), where we increase the iteration number $M$ in order to solve the larger instance size. The DT models as described and implemented by Bansal et al. [2] have training behavior which becomes increasingly unpredictable as the width $w$ (number of channels in the scratchpad) is reduced. This can be seen in Figure 3, where the variation in training loss at the end of training over different runs is larger in models of smaller width. It is also worth noting that increasing the width can result in explosive behavior in loss, including not-a-number (NaN) results, which can be seen in Figure D4.

## 3.2 Extrapolation Performance

The property of the DT networks that excited our interest was their ability to solve large instances than they were trained (that is, their extrapolative ability). This generalization to out-of-distribution examples is a key challenge for many machine learning algorithms. However, although DT-R can find models that extrapolate, very often the models that are trained extrapolate poorly. We observe models trained on few recurrences may have explosive behavior which only becomes a problem with extrapolative tasks requiring more recurrences.

Both DT and DT-R network used no bias terms in their model [2, 18]. In experimenting with these networks, we found that adding bias made these models very unstable. Unfortunately, having no biases exacerbates the 'dead cell problem', where some of the convolutional filters would have a zero response to all training inputs.

## 4 Refinement of the Architecture: Deep Thinking with Lipschitz Constraints

In the previous section we have identified a problem faced by DT-R, namely that there is instability in reaching a solution through the iteration in Equation (2). To overcome this difficulty we use a well known property of contraction maps. If $c : \mathcal{V} \to \mathcal{V}$ is a mapping of objects, $\boldsymbol{v}$ and $\boldsymbol{w}$, in a normed vector space $\mathcal{V}$, that is contractive in the sense that

$$\|c(\boldsymbol{v}) - c(\boldsymbol{w})\| < \|\boldsymbol{v} - \boldsymbol{w}\| , \tag{4}$$

implying $c(\cdot)$ is a Lipschitz function with Lipschitz constant $K < 1$, then the iterations

$$\boldsymbol{v}^{(m)} = c(\boldsymbol{v}^{(m-1)}) \tag{5}$$

will converge to a unique solution as a consequence of the Banach fixed-point theorem [1, 3]. We can use this to refine DT-R by engineering the network $\mathcal{G}(\cdot, \boldsymbol{x})$ to be a contraction mapping. Noting that in Equation (2) the input $\boldsymbol{x}$ is held constant throughout the iteration, we construct $\mathcal{G}(\cdot, \boldsymbol{x})$ so that it has an *approximate* Lipschitz constant $K$ less than 1.

Although any Lipschitz constant less than 1 would guarantee convergence, the nature of the problem solving mechanism we seek to learn intuitively means that we do not want fast convergence. This is because the network $\mathcal{G}$ performs local convolutions with a limited receptive field. However, finding good quality solutions often requires a global knowledge. To accumulate such knowledge requires long distance information to accumulate in the scratchpad vector $\phi^{(m)}$ over multiple iterations. To allow this to happen we choose the approximate Lipschitz constant associated with $\mathcal{G}$ to be less than, but close to 1.

## 4.1 Constraining the Lipschitz Constant

The primary tool we use to control the Lipschitz constant of $\mathcal{G}(\cdot, \boldsymbol{x})$ is spectral normalization [15]. This is an extension of spectral norm regularization, introduced by Yoshida and Miyato [20], but it sets the spectral norm of an operator to a fixed value rather than penalizing the norm of an operator that differs from a predefined value. The method uses a power iteration to compute an approximation of the spectral norm of an operator, updated after each gradient step. Computation of the spectral norm is a costly operation, but it only has to be done at the beginning of the iterative process.

As network $\mathcal{G}$ consists of convolutions we divide each convolution kernel weight by the spectral norm plus a small constant $\varepsilon$ to ensure the spectral norm is less than one; there is one exception in that the convolution applied to the recall connection $\boldsymbol{x}$ does not need normalizing — please see Appendix B for full details. In addition to using the spectral norm we need to make sure all the other transformations carried out by $\mathcal{G}$ are at most 1-Lipschitz. However, to avoid applying constraints to the constant recall connection, we replace DT-R's method of applying a single convolution on the concatenation $\left[\phi^{(m)}, \boldsymbol{x}\right]$ with two separate convolutional layers on $\phi^{(m)}$ and $\boldsymbol{x}$ respectively whose output is combined by element-wise addition. See Appendix A for a rigorous discussion on how the Lipschitz behaviour we require is ensured.

**Constraining Activation Functions.** To guarantee convergence under the constraints provided, any activation function used in $\mathcal{G}$ must be 1-Lipschitz. This includes Rectified Linear Units (ReLUs), Exponential Linear Units (ELUs) [4], and $\tanh$, but excludes Gaussian Error Linear Units (GELUs) [8], which have an absolute gradient greater than 1 around $x = \sqrt{2}$. Results in the body of this paper use the ELU activation for DT-L networks and the ReLU for DT-R as originally defined. Additional results using ReLU for DT-L and ELU for DT-R can be found in Appendix D.2.

**Constraining Residual Skip Connections.** DT and DT-R use element-wise addition for residual connections in recurrent blocks. The sum of two functions $\mathrm{c} : X \to Y$ and $\mathrm{d} : X \to Y$ applied to the same input, with Lipschitz constants $K_{\mathrm{c}}$ and $K_{\mathrm{d}}$ respectively results in the upper bound,

$$\|(\mathrm{c}(\boldsymbol{x}_1) + \mathrm{d}(\boldsymbol{x}_1)) - (\mathrm{c}(\boldsymbol{x}_2) + \mathrm{d}(\boldsymbol{x}_2))\| \leq (K_{\mathrm{c}} + K_{\mathrm{d}})\|\boldsymbol{x}_1 - \boldsymbol{x}_2\| \tag{6}$$

for all $\boldsymbol{x}_1, \boldsymbol{x}_2 \in X$. In the case of a standard residual connection, one function is the identity $(\mathrm{id}(\boldsymbol{x}) = \boldsymbol{x})$ and the other is the block of layers $\mathcal{B}$ contained in the span of the residual connection. Under our constraints, the identity is 1-Lipschitz and the block of layers is $K_{\mathcal{B}}$-Lipschitz, where $K_{\mathcal{B}} \in [0, 1)$. This results in a Lipschitz upper bound for the output of $1 + K_{\mathcal{B}}$.

As a result, residual connections as addition between the identity and a block of layers can increase the Lipschitz constant of the recurrent part even if the layers themselves are 1-Lipschitz. A solution to this, which also allows more expression in the model, is to make each residual connection a parametric linear interpolation between the identity output and the block output

$$(1 - \gamma)\,\mathrm{id}(\boldsymbol{x}) + \gamma\,\mathcal{B}(\boldsymbol{x}), \quad \gamma \in [0, 1] \,. \tag{7}$$

DT-L applies this interpolation to each channel, $c$, individually; an unconstrained learnable parameter $\bar{\gamma}_c$ exists for each channel in each residual connection, then the residual parameters are set to $\gamma_c = \sigma(\bar{\gamma}_c)$ where $\sigma$ is the logistic function to ensure that $0 \leq \gamma_c \leq 1$.

## 4.2 Additional Modifications

The above changes leads to a more stable network, allowing us to make additional modifications and still obtain convergent behaviour with improved performance. Without explicitly controlling the Lipschitz constant, these changes often lead to complete failure of the network to solve the problem. In our final model design we apply three extra modifications that lead to consistent improvements:

1. Batch normalization layers [10] are added for the input and after each convolution, except for layers in the recurrent block and the final output layer. Empirically this improves performance.

2. A bias term is added to the recall convolution (without the Lipschitz constraint) in $\mathcal{G}$. The bias term is added to the recall convolution to mitigate the dead cell problem. Since this is a constant addition (along with recall) it does not disrupt the convergence constraints we have put in place (see Appendix A). A bias term is also added to the final (output) layer and, when batch normalization is not used, to $\mathcal{F}$.

3. Exponential Linear Unit (ELU) activations [4] are used instead of rectified linear unit (ReLU) activations. This choice is influenced by the desire to promote activations staying close to zero [4] throughout iteration, as well as mitigating the effect of the dying ReLU problem [19] where 'dead' activations increase with depth [13]. Artificially deep models like DT are therefore likely to encounter this problem. Using ReLU can also result in a rapid rate of convergence, preventing the model from learning complex algorithms for harder tasks (see Appendix D.2).

Our reasoning for not applying batch normalization in the recurrent block follows from Jastrzebski et al. [11] where unsharing batch normalization statistics was necessary to avoid activations exploding. In an architecture where the maximum number of iterations is unknown it appears infeasible to unshare batch normalization statistics for every iteration.

Using these modifications — creating a model we call Deep Thinking with Lipschitz Constraints (DT-L) — greatly improves the stability of training and using these networks. As we will see in Section 5, the new DT-L network allow us to solve the same problems explored by DT and DT-R, but using, in some cases, three orders of magnitude less parameters while achieving high performance and a more consistent success rate during training. The increased reliability offered by DT-L allows us to explore many other modifications enabling us to tackle more sophisticated problems. We illustrate this by attempting to find good solutions to one of the most notoriously difficult problems, namely, the traveling salesperson problem (TSP).

## 5  Results on Easy-to-Hard Problems

In this section, we compare our model (DT-L) against DT-R. Note that DT-R is an improvement of DT by the same authors and has already been shown to have better performance [2]. We test on the three problem classes used by Bansal et al. [2] to evaluate DT-R, namely a **prefix sum problem**, a **maze problem** and a **chess problem**.

We have trained and evaluated the models on a range of different Nvidia GPU accelerators from RTX2080Tis to A100s, as well as on M3-series Apple Silicon. Memory usage is insignificant compared to available GPU memory. Training time is of the order of 30 minutes for the $w = 32$ model on the prefix-sum problem using a single RTX8000. More details can be found in Appendix E, and code for the experiments can be found at `https://github.com/Jay-Bear/rethinking-deep-thinking`.

The accuracy given in this section measures the proportion of instances where the network produces the exact correct solution. Note that even if a predicted solution differs from the target by one element (e.g. 1 bit in prefix sums, 1 pixel in mazes) it is considered a failure.

**Prefix Sums.**   The prefix sum problem involves translating a string of ones and zero to a new string that counts sequences of ones (details are described by Bansal et al. [2]). The problem is simpler than the the maze and chess problem, and considerably faster to run. As a consequence we choose this as the main problem to perform a comparative study.

In our experiments we compare the DT-R model to our DT-L model. Both were trained on instances consisting of 32 bits for a maximum of 30 iterations, using IPT with $\alpha = 0.5$. Both models have a width (number of channels) of $w = 32$ (more details about architecture can be found in Appendix B.1). We found that for this problem there is little change in performance above $w = 32$ for DT-L. In Figure 4 we show the solution accuracy for 30 randomly-initialized models of both DT-R and DT-L on the 512-bit test dataset versus the number of iterations ($M$) to generate those solutions. We measured the performance on 10 000 randomly generated instances of each problem. The error in

the mean is less than 0.5% which is approximately the width of the lines in the figure. As can be seen DT-R struggles to consistently find networks that extrapolate to larger problem settings (it only obtains greater than 90% accuracy on two out of 30 training runs). In contrast DT-L only fails to reach above 90% accuracy on two out of 30 training runs.

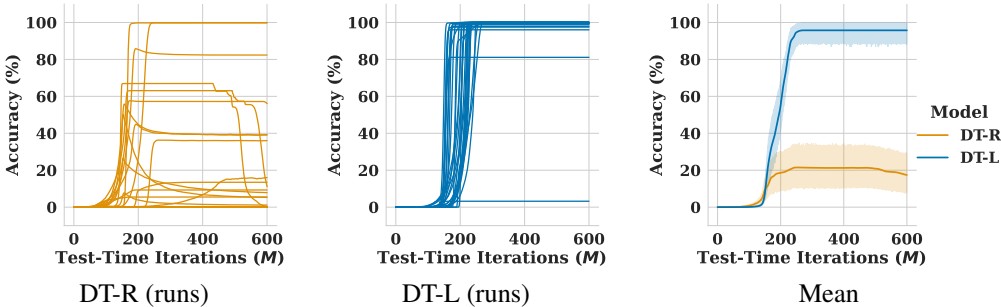

| DT-R (runs) | DT-L (runs) | Mean |

Figure 4: Comparison between Deep Thinking with Recall and Deep Thinking with Lipschitz Constraints on the prefix sums problem. Two left plots show the solution accuracy of inference-time runs on 512-bit problems for 30 individual models each. Each line corresponds to the performance of a network trained from scratch with different randomly initial weights. The accuracy is measured on 10 000 problem instances. The right plot shows the mean of all 30 for each. Models have a channel width of $w = 32$. Shaded areas show 95% confidence intervals.

To emphasize DT-L's performance on smaller widths, we selected $w = 32$ as the primary width for comparison. We perform ablations on other aspects of the model in Appendix D.2.

**Mazes.** The mazes problem consists of drawing the correct path for a blank maze, given a starting point and an ending point. Details about the representation of mazes is given in Schwarzschild et al. [17] and Bansal et al. [2]. Our tests are performed on models trained on $17 \times 17$ mazes, extrapolated to $33 \times 33$ mazes. Results from multiple runs and an aggregate can be seen in Figure 5. The maze models defined by Bansal et al. [2] are trained on $9 \times 9$ mazes. Attempting to train DT-L on mazes of this size resulted in the models often learning trivial solutions which did not extrapolate to larger mazes.

**Chess Puzzles.** The chess puzzles problem involves identifying the next best move of the current board state. Specifically, the model must learn to classify 1s in two cells representing the piece to move and the location to move to, and 0s elsewhere. The models are trained on a train/validation split of the easiest 600,000 problems, where difficulty is based on Lichess.org rankings (see Schwarzschild et al. [17] for more detail).

This problem in particular can be viewed as different to both the prefix sums and mazes problems in that the difficulty of the problem doesn't come from the size. All chess puzzles in this dataset are standard $8 \times 8$ chess boards. For all tests on the chess dataset [17] we follow the same number of epochs as Bansal et al. [2]. Results are shown in Figure 6. DT-L achieves very similar performance to DT-R, seemingly hitting the same apparent ceiling on performance discussed by Bansal et al. [2]. We would like to explore the reasons for this in more detail the future, but we note that at training time the models do not reach 100% accuracy, so it should not be a surprise that they do not always extrapolate. It is possible that the issue arises from the structure of the data and the way it interacts with the model architecture.

## 6 Benchmarking on Traveling Salesperson

To demonstrate the strength of DT-L, we have formulated the traveling salesperson problem as a differentiable optimization task for the model to solve. This is a significantly more challenging problem than those discussed in Section 5. The problem instance $x$ is now the matrices of "distances" with additional information. For arbitrary distance matrices this is known to be NP-hard. An optimal solution corresponds to a tour (permutation of the cities) such that the sum of distances between neighboring cities is minimized, although we seek only to find a low cost tour.

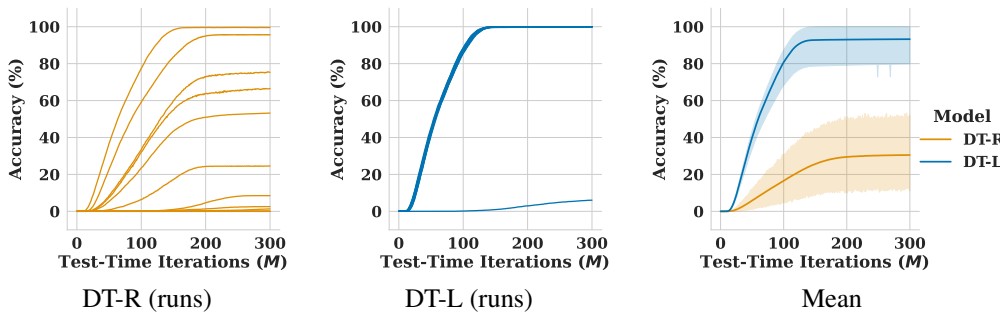

| DT-R (runs) | DT-L (runs) | Mean |

Figure 5: Comparison between Deep Thinking with Recall and Deep Thinking with Lipschitz Constraints on the mazes problem for small models. Two left plots show the solution accuracy of inference-time runs on $33 \times 33$ mazes for 14 different models each. The right plot shows the mean of all 14 for each. Models have a channel width of $w = 32$. Shaded areas show 95% confidence intervals.

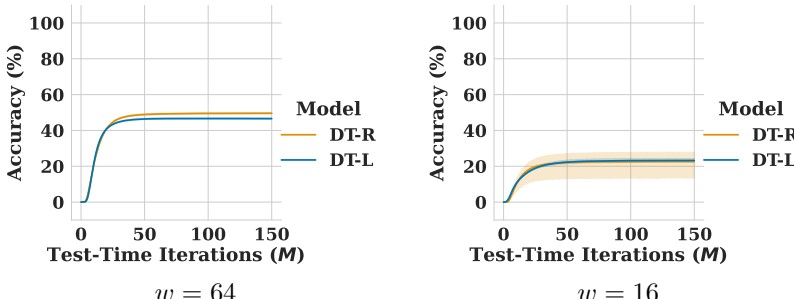

| $w = 64$ | $w = 16$ |

Figure 6: Comparison between Deep Thinking with Recall and Deep Thinking with Lipschitz Constraints on the chess puzzles problem on models of two different widths, showing the mean accuracy of inference-time runs on problems ranked in increasing difficulty between 1,000,000 and 1,100,000. Aggregate of six models, each trained on the easiest 600,000 problems, where shaded areas show 95% confidence intervals.

The solution vector $\boldsymbol{y}$ is a binary matrix with 1s corresponding to the edges that are used in the tour. We constructed the network so that the solution vector would correspond to a feasible solution. Unfortunately, in doing so involves an intermediate representation where the evaluation of the cost is non-trivial, which made the problem of learning an iterative solution very challenging. To overcome this, we increased the expressiveness of the scratchpad vectors $\phi^{(m)}$ to learn orthogonal transforms that it applied to a part of the scratchpad vector. This considerably improved the performance of the network. Details of the modification used are given in Appendix B.1.

## 6.1 TSP Results

In Table 1 we give the mean tour length for randomly generated instances from two class of problems: symmetric random tours and asymmetric random tours. The DT-L models were trained for each problem class on tours of size $n = 15$ with $M = 45$. We show both interpolative results (using a new set of tours on the same size problems) and extrapolative results where we test of problems of size $n = 30$ with $M = 120$. For comparison we provide the length of random tours, the length of tours generated by the greedy nearest neighbor (NN) algorithm, and the length of tours from a modified version of the NN algorithm which selects the lowest-cost NN tour out of all starting points instead of starting from a random point (BNN).

The results in Table 1 show DT-L's ability to learn non-trivial algorithms by performing considerably better than random tours. That the results are worse than the nearest neighbor methods should not be surprising, as in the asymmetric non-Euclidean case these are amongst the best known algorithms.

The models take approximately 8 hours to train on an RTX2080Ti GPU for 80,000 batches of 64 randomly generated TSPs for both symmetric and asymmetric settings with $n = 15$ and $M = 45$.

Testing was performed on M3 Apple Silicon with MPS acceleration and takes 3.59ms per $n = 15$ problem instance for $M = 45$ and 23ms per $n = 30$ problem with $M = 120$.

Table 1: Results for TSP runs on symmetric and asymmetric instances. The results are the mean for 12,800 random instances. For DT-L, $M = 45$ was used for problem size $n = 15$ and $M = 120$ for $n = 30$, where tour lengths are shown in column 2. Column 3 (Random Tours) gives the tour length for random tours, while columns 4 and 5 give the tour lengths for greedy nearest neighbor tours starting from a random point (NN) or choosing the lowest NN tour from all starting points (BNN) respectively.

| Problem | DT-L | Random Tours | NN Tours | BNN Tours |
|---|---|---|---|---|
| Symmetric $n = 15$ | $3.99 \pm 0.006$ | $7.5 \pm 0.01$ | $2.85 \pm 0.005$ | $2.31 \pm 0.005$ |
| Symmetric $n = 30$ | $6.02 \pm 0.007$ | $15.0 \pm 0.01$ | $3.50 \pm 0.005$ | $2.72 \pm 0.004$ |
| Asymmetric $n = 15$ | $4.66 \pm 0.008$ | $7.5 \pm 0.01$ | $2.82 \pm 0.006$ | $2.09 \pm 0.004$ |
| Asymmetric $n = 30$ | $7.7 \pm 0.01$ | $15.0 \pm 0.01$ | $3.50 \pm 0.006$ | $2.53 \pm 0.004$ |

## 7 Discussion

Deep Thinking-style architectures provide a new paradigm for using machine learning for general problem solving. The key idea is to use a recurrent architecture to find a solution through multiple iterations. This paper has addressed a major drawback of the published models which is the instability that frequently arises in the iteration steps. Having addressed these problems not only are we able to obtain networks that much more reliably solve new problems, but we show that we can run much smaller networks with similar and often better performance. To illustrate the potential of this approach we have tackled a notoriously difficult problem, namely TSP.

Deep Thinking approaches are attractive because they learn a general problem solving strategy that can be used to solve considerably larger instances of the problem than they were trained on. Extrapolation to problems outside the training dataset is an area where traditional machine learning struggles. Clearly, the problems these models extrapolate to are in the same problem class to the training data, but it is noteworthy that the strategies learned for the problem classes we have investigated scale in this way. Another interesting feature is that the network learns the problem solving strategy through examples (in the case of TSP it is not even shown any low cost solutions). Admittedly, for TSP we needed to construct a network that outputted tours and to get the quality of results we did we introduced a mechanism for learning orthogonal transformation which we then applied to part of the scratchpad vector. However, we did not build in any explicit rules for solving the TSP — so much so that we do not fully understand the algorithm DT-L uses to find low cost tours, particularly in the case of asymmetric non-Euclidean TSP.

**Broader impact.** Given the ubiquity of iterative algorithms in solving problems, having a mechanism to learn such algorithms opens up a lot of possibilities. The advantage of using a recurrent mechanism over a feed-forward architecture is that to solve a larger problem we simply needed to run the recurrent loop more often; this is often desirable in the real world because it might be impossible to train on sufficient data that you can guarantee not to need to extrapolate at inference time. Clearly, models of this kind have very many potential applications - both in developing *new* (and potentially improved — could we learn an algorithm that is significantly faster or more energy efficient than anything currently existing?) algorithms for existing problem classes where we already have solutions, as well as in all areas of predictive modelling where we believe the input-output relationship is best captured through a potentially-deep recurrent function. Inevitably, some of the potential uses also have the potential for misuse.

**Limitations.** This work shows theoretically and empirically that it is possible to modify an architecture to learn recurrent functions with convergence guarantees. The problems we solve are relatively simple however. The performance we obtain, for example, on TSP is far from state-of-the-art, but we believe that the contribution is significant as we are able to find a heuristic algorithm running in $M = O(n \log(n))$ steps through trial and error, without any explicit inductive bias towards finding what we might consider to be a sensible algorithm. There are also clearly other potential issues with the approach; learnability is significantly improved over previous attempts, but there are still cases

where the model fails to learn. The use of convolutions means that the only way to capture long-term (or large distance) dependencies in the data is through iteration; this is possibly a benefit, but other architectures with different structural biases might work better on certain problems. We believe, in terms of comparison to NN and BNN heuristics, this local view of the distance matrix may contribute to the disappointing appearance of the results.

**Outlook.** This work leaves many open directions for future research. It would be interesting to explore different architectures, such as Transformers, and more diverse problem settings. It would also be fascinating to better understand what algorithms are learned by the model under different settings and whether for example the complexity of the learned algorithm can be controlled.

## Acknowledgments and Disclosure of Funding

JB is funded by a PhD studentship graciously provided by the School of Electronics and Computer Science. JH was supported by the Engineering and Physical Sciences Research Council (EPSRC) International Centre for Spatial Computational Learning [EP/S030069/1]. The authors acknowledge the use of the IRIDIS X High Performance Computing Facility, and the Southampton-Wolfson AI Research Machine (SWARM) GPU cluster generously funded by the Wolfson Foundation, together with the associated support services at the University of Southampton in the completion of this work.

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

# A  On Being Lipschitz

As discussed in the main paper one of its major contribution is the observation that by making $\mathcal{G}(\cdot, \boldsymbol{x})$ $K$-Lipschitz with $K < 1$, then the iterative mapping defined in Equation (2) is guaranteed to converge to a unique solution as a consequence of the Banach fixed-point theorem [1]. To engineer $\mathcal{G}(\cdot, \boldsymbol{x})$ to be $K$-Lipschitz we rely on a few well known properties of functions. If $\mathcal{C} : X \rightarrow Z$ is a composition of two Lipschitz mappings $\mathcal{A} : X \rightarrow Y$ and $\mathcal{B} : Y \rightarrow Z$ with Lipschitz constant $K_A$ and $K_B$ then

$$\|\mathcal{C}(\boldsymbol{x}) - \mathcal{C}(\boldsymbol{y})\| = \|\mathcal{B}(\mathcal{A}(\boldsymbol{x})) - \mathcal{B}(\mathcal{A}(\boldsymbol{y}))\| \leq K_B \|\mathcal{A}(\boldsymbol{x}) - \mathcal{A}(\boldsymbol{y})\| \leq K_A K_B \|\boldsymbol{x} - \boldsymbol{y}\|. \quad \text{(A1)}$$

Thus a sufficient condition to ensure $\mathcal{C}$ is $K$-Lipschitz with $K < 1$ is that $K_A, K_B < 1$. This trivially generalises to any number of compositions.

In the network $\mathcal{G}(\boldsymbol{\phi}, \boldsymbol{x})$ the input vector $\boldsymbol{x}$ is added to $\boldsymbol{\phi}$ through a convolution. However, adding an offset vector to a mapping does not affect the Lipschitz property as if $\mathcal{A}$ is K-Lipschitz and $\boldsymbol{c}$ is a constant vector, then the mapping $\mathcal{B}(\boldsymbol{x}) = \mathcal{A}(\boldsymbol{x}) + \boldsymbol{c}$ satisfies

$$\|\mathcal{B}(\boldsymbol{x}) - \mathcal{B}(\boldsymbol{y})\| = \|\mathcal{A}(\boldsymbol{x}) - \mathcal{A}(\boldsymbol{y})\| \leq K \|\boldsymbol{x} - \boldsymbol{y}\|. \quad \text{(A2)}$$

Since during every iteration the input vector $\boldsymbol{x}$ and the convolution filters remain unchanged, the addition of the input $\boldsymbol{x}$ (which the Deep Thinking with Recall authors termed the recall) does not change the Lipschitz property of the network.

Finally, to ensure the Lipschitz behaviour of the convolutions we note that convolutions are equivalent to applying a matrix, $\mathbf{C}$, to some vector, $\boldsymbol{v}$. For a matrix norm $\|\mathbf{C}\|$ that is compatible with a vector norm $\|\boldsymbol{x}\|$,

$$\|\mathbf{C}\,\boldsymbol{v}\| \leq \|\mathbf{C}\|\,\|\boldsymbol{x}\|, \quad \text{(A3)}$$

where there exists a vector $\boldsymbol{x}$ where the equality conditions holds. If $N = \|C\|$ then, through the linearity of norms, $\mathbf{D} = (K/N)\,\mathbf{C}$ will have norm $K/N$ so that the linear mapping $\mathcal{D}(\boldsymbol{x}) = \mathbf{D}\,\boldsymbol{x}$ is $K$-Lipschitz since

$$\|\mathcal{D}(\boldsymbol{x}) - \mathcal{D}(\boldsymbol{y})\| = \|\mathbf{D}(\boldsymbol{x} - \boldsymbol{y})\| = \frac{K}{N}\|\mathbf{C}(\boldsymbol{x} - \boldsymbol{y})\| \leq \frac{K}{N}\|\mathbf{C}\|\,\|\boldsymbol{x} - \boldsymbol{y}\| = K\,\|\boldsymbol{x} - \boldsymbol{y}\|. \quad \text{(A4)}$$

This would be true for any normed vector space with an appropriate compatible matrix norm. In this paper we have used the $\ell_2$ norm where the compatible matrix norm is the spectral norm (i.e. the largest singular value).

To allow the algorithm being run by network $\mathcal{G}$ to accumulate enough information to find a good solution we want the Lipschitz constant, $K$, of network $\mathcal{G}$ to be as close to 1 as possible. If the network can find a good solution rapidly then it can find a scratchpad vector $\boldsymbol{\phi}$ in a region where the effective Lipschitz constant is less than 1. Indeed, we observed that despite Deep Thinking with Recall most often having Lipschitz constant greater than 1, nevertheless in the few runs where a successful network was learned, the growth in the solution $\|\boldsymbol{\phi}^{(m+1)}\|/\|\boldsymbol{\phi}^{(m)}\|$ could be less than 1. Being on the edge of the stable region seems to be helpful for the network to find good solutions to hard problems. Consequently, in engineering the network $\mathcal{G}$ we attempt to make the Lipschitz constant for most mapping layers as close to 1 as possible.

# B  Architecture

In this section we describe the architecture of DT [17], DT-R [2] and our network DT-L. For DT and DT-R the architecture is identical to that described in the original paper. We include a description here to make the comparison with DT-L clearer. The number of channels of the input, $\boldsymbol{x}$, varies depending on the problem class. It is 1 for prefix sums, 3 for mazes, 12 for chess and 3 for TSP. Following DT and DT-R, the output vector, $\boldsymbol{y}$, has two channels. During training this is used to compute a categorical cross-entropy loss, while at inference time this is put into a max function to obtain a binary output.

In all models the convolutions are either 3 (for prefix sum) or $3 \times 3$ with stride 1 and padding 1. In the figures below the boxes with rounded edges represent convolutions.

**Deep Thinking.** In Figure B1 we show the Deep Thinking (DT) model. Note in this case the input vector $x$ is not given to the network at each iteration of $\mathcal{G}$.

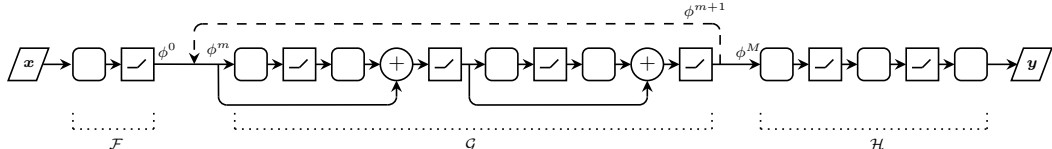

Figure B1: The DT architecture of Schwarzschild et al. [18]. Arrows denote flow from one function to the next with recurrent iterations are denoted by the dashed arrow. Single-lined rounded rectangles denote regular convolutions. Activation functions in square boxes are ReLU.

**Deep Thinking with Recall.** In Figure B2 we show the architecture for the Deep Thinking with Recall (DT-R) model. In DT-R the network is fed $x$ every iteration and it is concatenated to the recurrent connection.

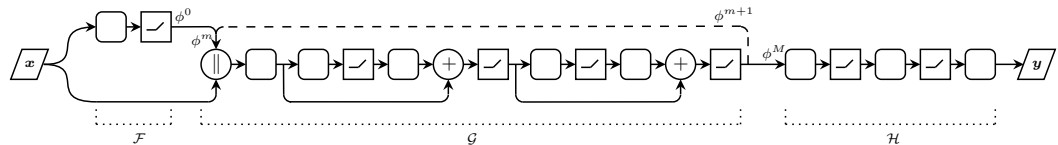

Figure B2: The DT-R architecture of Bansal et al. [2]. Arrows denote flow from one function to the next with recurrent iterations are denoted by the dashed arrow. Single-lined rounded rectangles denote regular convolutions. The recall $x$ is concatenated ($\|$) to the recurrent input. Activation functions in square boxes are ReLU by default, but we experiment with ELU in Appendix D.2.

**Deep Thinking with Lipschitz Constraints.** The architecture of our model, Deep Thinking with Lipschitz Constraints (DTL) is shown in Figure B3. The convolutions with spectral norms are shown in bold. Following common practice when working with models with skip connections we have also added batch norms (shown as solid black rectangles). These ensure that the tensors going through the network are normalized ensuring that their means are around zero where the non-linearity in the activation functions are strongest. Empirically this lead to improved performance as shown by ablation studies.

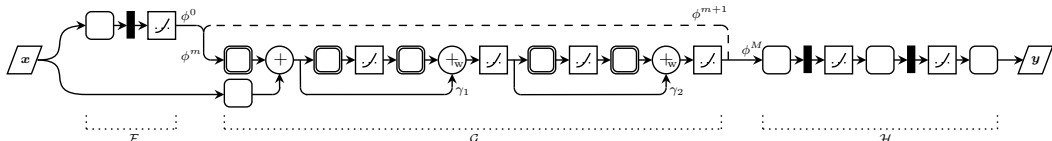

Figure B3: Our basic DT-L architecture. Arrows denote flow from one function to the next with recurrent iterations are denoted by the dashed arrow. Double-lined rounded rectangles denote convolutions with the Lipschitz constraint applied. Single-lined rounded rectangles are regular convolutions. Solid black rectangles denote batch-normalization operations. Activation functions are ELU by default, but we experiment with ReLU in Appendix D.2. Plus symbols with a small w indicate weighted summation, with the residual weighted by $1 - \gamma$ and main branch by $\gamma$ (c.f. Equation (7)). Rather than concatenating the recall $x$ into the recurrent input we pre-process it with a convolution and then sum with a convolved version of the recurrent input.

## B.1 TSP Model

Recall that in TSP the input vector $x$ corresponds to the matrices of distances[2]. The solution $y$ is a binary matrix showing which edges we used in the tour. Thus, the cost of the full tour is $\text{sum}(x \odot y)$,

---

[2]Technically these are pseudo-distances as although they are positive they are not necessarily symmetric and are not guaranteed to satisfy the triangular inequality.

where $\odot$ denotes element-wise multiplication. This is just the sum the set of edge distances that are used in the tour. This is schematically shown below.

$$
\underbrace{\begin{pmatrix}
0.00 & 0.33 & 0.78 & 0.22 & 0.57 & 0.25 \\
0.75 & 0.00 & 0.49 & 0.22 & 0.39 & 0.49 \\
0.83 & 0.31 & 0.00 & 0.28 & 0.61 & 0.56 \\
0.37 & 0.18 & 0.50 & 0.00 & 0.78 & 0.13 \\
0.19 & 0.61 & 0.11 & 0.68 & 0.00 & 0.49 \\
0.30 & 0.64 & 0.82 & 0.71 & 0.68 & 0.00
\end{pmatrix}}_{\boldsymbol{x}}
\underbrace{\odot}_{\odot}
\underbrace{\begin{pmatrix}
0 & 0 & 0 & 1 & 0 & 0 \\
0 & 0 & 0 & 0 & 1 & 0 \\
0 & 1 & 0 & 0 & 0 & 0 \\
0 & 0 & 0 & 0 & 0 & 1 \\
0 & 0 & 1 & 0 & 0 & 0 \\
1 & 0 & 0 & 0 & 0 & 0
\end{pmatrix}}_{\boldsymbol{y}}
=
\begin{pmatrix}
0.00 & 0.00 & 0.00 & 0.22 & 0.00 & 0.00 \\
0.00 & 0.00 & 0.00 & 0.00 & 0.39 & 0.00 \\
0.00 & 0.31 & 0.00 & 0.00 & 0.00 & 0.00 \\
0.00 & 0.00 & 0.00 & 0.00 & 0.00 & 0.13 \\
0.00 & 0.00 & 0.11 & 0.00 & 0.00 & 0.00 \\
0.30 & 0.00 & 0.00 & 0.00 & 0.00 & 0.00
\end{pmatrix}
$$

The edge matrix $\boldsymbol{y}$ is a matrix with a 1 in each row and each column and zero elsewhere. That is, it is a permutation matrix. Unfortunately, not all permutation matrices correspond to legal tours. The only permutations that are allowed correspond to irreducible permutation (i.e. those consisting of a single cycle of length $n$, where $n$ is the number of cities)—these are cyclic permutations of order $n$. This, arises because there are $(n-1)!$ tours (since the starting city is arbitrary) whereas there are $n!$ permutation matrices. The illegal permutations consist of multiple cycles (e.g. for a four city problem we might have $1 \leftrightarrow 2$ and $3 \leftrightarrow 4$.). It is however challenging to generate irreducible permutations. To solve this problem we use the fact that the set of permutations of order $n$ form an equivalence class $[n]$ and for any permutation $\boldsymbol{q} \in S_n$ (where $S_n$ is the group of all permutations of $n$ objects—known as the symmetric group) then if $\boldsymbol{\pi} \in [n]$ the conjugate $\boldsymbol{q}\,\boldsymbol{\pi}\,\boldsymbol{q}^{-1} \in [n]$. This is a well known result in group theory, but it means that if we start with an irreducible representation $\boldsymbol{\pi}$ (e.g. the tour $1 \to 2 \to 3 \to \cdots \to n \to 1$) then for any permutation matrix $\boldsymbol{q}$ the product $\boldsymbol{q}\,\boldsymbol{\pi}\,\boldsymbol{q}^{-1}$ represents a legal tour. Thus we set up the networks to generate a permutation matrix $\boldsymbol{q}$. This used a Gumbel-Sinkhorn network [14] as part of network $\mathcal{H}$. Occasionally this would generate a solution with fractional edges. To avoid this we added a term to the loss function that punished solutions with fractional edges.

Although this procedure guarantees that the solutions correspond to feasible tours, the quality of tours obtained when we trained the full network is poor. The problem is that the relationship between the permutations $\boldsymbol{q}$ and the matrices of edges visited $\boldsymbol{y} = \boldsymbol{q}\,\boldsymbol{\pi}\,\boldsymbol{q}^{-1}$ is difficult to learn. To address this problem in the scratch-pad vector we learned three groups of channels $\phi = (\phi_1, \phi_2, \boldsymbol{m})$. The last part $\boldsymbol{m}$ is a single channel which we treat as a matrix from which we compute an orthogonal transformation. To achieve this we perform SVD to obtain $\boldsymbol{u}\boldsymbol{s}\boldsymbol{v}^{\mathsf{T}}$. We discard $\boldsymbol{s}$ to obtain an orthogonal matrix $\boldsymbol{w} = \boldsymbol{u}\boldsymbol{v}^{\mathsf{T}}$. We use this to transform $\phi_2$. This reordering significantly improves the quality of the tours we learn. We attribute this to providing the network $\mathcal{G}$ with the ability to understand how altering $\boldsymbol{q}$ will change $\boldsymbol{y} = \boldsymbol{q}\,\boldsymbol{\pi}\,\boldsymbol{q}^{-1}$ and hence the cost of a tour. We used a differentiable version of SVD from the PyTorch library [16]. Very occasionally this would fail, in which case we terminated the training and start again. Although rare this was the one source of failure of the network.

For asymmetric tours we obtained improved results by making the input $\boldsymbol{x}$ have three channels. The first being the distance matrix and the second the transpose of the distance matrix and the third a matrix that have values of 1 in the upper triangle, 0 along the diagonal and -1 in the low triangle. For convenience the same input was used for symmetric instances, but obviously the transposed distance matrix is equal to the distance matrix. In addition, the scratch-pad vector learnt two matrices $\phi = (\phi_1, \phi_2, \boldsymbol{m}_1, \boldsymbol{m}_2)$ from which we constructed two orthogonal matrices $\boldsymbol{w}_i = \boldsymbol{u}_i \boldsymbol{v}_i^{\mathsf{T}}$ which we used to apply both column-wise and row-wise transformations $\boldsymbol{w}_1 \phi_2 \boldsymbol{w}_2^{\mathsf{T}}$.

With these modifications we then trained DT-L on randomly generated instances of TSP using a loss function based on the mean cost of a single edge in the selected tour. The distances were normalized so the maximum edge is 1. The loss function can be stated as

$$
\mathcal{L}_{\text{tour}} = \frac{1}{N} \sum_{ij} \left[ (x_{ij} - 1) \cdot y_{ij}^{\xi} \right] \tag{B5}
$$

where $\xi \geq 1$ is a hyperparameter that pushes values not equal to 1 closer to zero, thus preventing solutions with fractional edges.

**Prefix Sums Models**

For different widths $w$, we scale the channels of the convolutional layers in $\mathcal{H}$ (output module) as follows:

| Layer | Input Channels | Output Channels |
|---|---|---|
| Convolution 1 | $w$ | $w$ |
| Convolution 2 | $w$ | $\max(2, \lfloor w/2 \rfloor)$ |
| Convolution 3 | $\max(2, \lfloor w/2 \rfloor)$ | 2 |

This ensures consistency with the $w = 400$ prefix sums models in [2]. For $w = 32$, this results in output channels of 32, 16, and 2.

**Mazes Models**

For different widths $w$, we scale the channels of the convolutional layers in $\mathcal{H}$ (output module) as follows:

| Layer | Input Channels | Output Channels |
|---|---|---|
| Convolution 1 | $w$ | $\max(2, 4^{-1}w)$ |
| Convolution 2 | $\max(2, 4^{-1}w)$ | $\max(2, 4^{-2}w)$ |
| Convolution 3 | $\max(2, 4^{-2}w)$ | 2 |

**Chess Puzzles Models**

For different widths $w$, we scale the channels of the convolutional layers in $\mathcal{H}$ (output module) as follows:

| Layer | Input Channels | Output Channels |
|---|---|---|
| Convolution 1 | $w$ | $\max(2, 16^{-1}w)$ |
| Convolution 2 | $\max(2, 16^{-1}w)$ | 2 |
| Convolution 3 | 2 | 2 |

## C  Training

Unless specified otherwise:

- All models use the Adam optimizer [12] with
  - a learning rate of 0.001,
  - $\beta_1 = 0.9$, $\beta_2 = 0.999$,
  - weight decay set to 0.0002 and only applied to unconstrained convolutional weights;
- incremental progress training with $\alpha = 0.5$;
- exponential warmup with a warmup period of 3;
- a multi-step learning rate scheduler where milestones are calculated as a $8 : 4 : 2 : 1$ ratio of the total number of epochs, with learning rates multiplied by 0.1 at each milestone.

As an example, prefix-sums-solving models are trained to 150 epochs. With the given ratio this produces milestones of 80, 120, and 140.

For all models, the end-of-epoch state resulting in the best validation score (accuracy for non-TSP problems, loss for TSP) became the final result of training. This is standard practice in training deep learning models.

## C.1 Prefix Sums

Prefix-sums-solving models were trained on 32-bit prefix sums from the EasyToHard dataset [17] for 150 epochs, shuffled and split into 80% training samples, 20% validation samples. Each batch contained 500 samples.

Trained models for ablation studies instead used milestones at a $4 : 2 : 1$ ratio of 150 epochs instead.

Models were trained with $M = 30$.

## C.2 Mazes

Maze-solving models were trained on $17 \times 17$ mazes from the EasyToHard dataset [17] for 50 epochs, shuffled and split into 80% training samples, 20% validation samples. Each batch contained 50 samples.

Models were trained with $M = 30$.

## C.3 Chess Puzzles

Chess puzzles models were trained on the easiest 600,000 chess instances from the EasyToHard dataset [17] for 120 epochs, with batch sizes of 300 problem instances. Learning rate scheduling followed the same $8 : 4 : 2 : 1$ milestone ratio, with a multiplier of 0.1. The dataset was shuffled and split into 80% training samples, 20% validation samples. Each batch contained 300 samples.

Models were trained with $M = 30$.

## C.4 Traveling Salesperson

Models trained to produce tours for TSP instead use stochastic gradient descent (SGD) with a learning rate of 0.001, Nesterov momentum of 0.9, and a weight decay of 0.0002 (applied only to unconstrained convolutional weights).

The model trained to perform symmetric TSP used IPT with $\alpha = 0.5$, whereas the model trained to perform asymmetric TSP did not use IPT.

Since there is no dataset, we chose 1,000 samples to be an appropriate size for one epoch. With 80% training samples and 20% validation samples, this produces 800 training batches per epoch and 200 validation samples per epoch. We generate 64 grids of distances per batch.

# D   Additional Results and Analysis

## D.1 DT-R Stability

To show the training instability in more widths $w$ for the prefix sums DT-R models, Figure D4 shows extended results from Figure 3.

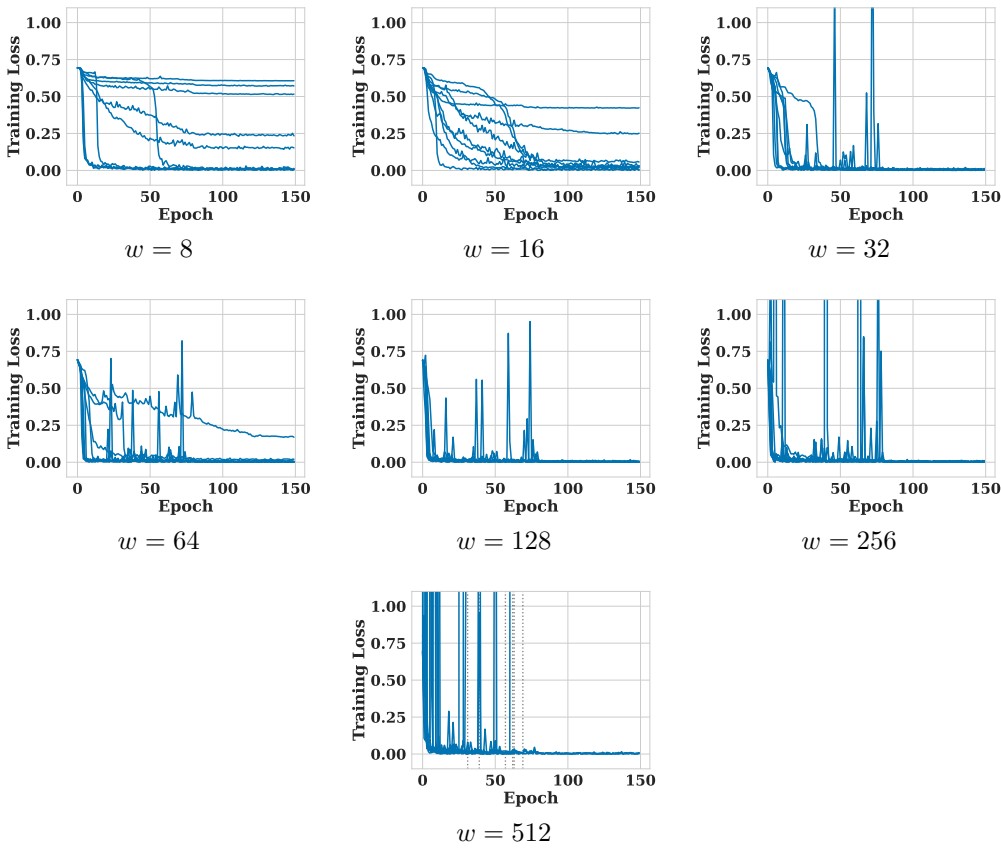

Figure D4: Extended results of Figure 3. Mean training (cross-entropy) loss at each epoch for prefix-sums-solving models of varying width $w$. Each curve is measured from a different random initialization of the model throughout training, for 10 models of each width. Dotted vertical lines indicate loss becoming NaN or infinite, where the model does not recover.

## D.2 Ablation Studies

We have performed ablation studies on prefix-sums-solving models of width $w = 32$ and using IPT with $\alpha = 1.0$. Instead of the $8 : 4 : 2 : 1$ milestone ratio (Appendix C), these models use a $4 : 2 : 1$ milestone ratio, meaning the multiplier $0.1$ is only applied twice instead of three times.

**Activation Function.** We perform a study by modifying DT-R to use ELU [4] instead of ReLU. From Figure D5(a) it appears that ELU allows for increased extrapolation performance from DT-R with minimal decay.

Similarly, we perform a study by modifying DT-L to use ReLU instead of ELU. It can be seen from Figure D5(b) that, while ReLU results in faster convergence, ELU appears to provide overall increased extrapolation performance.

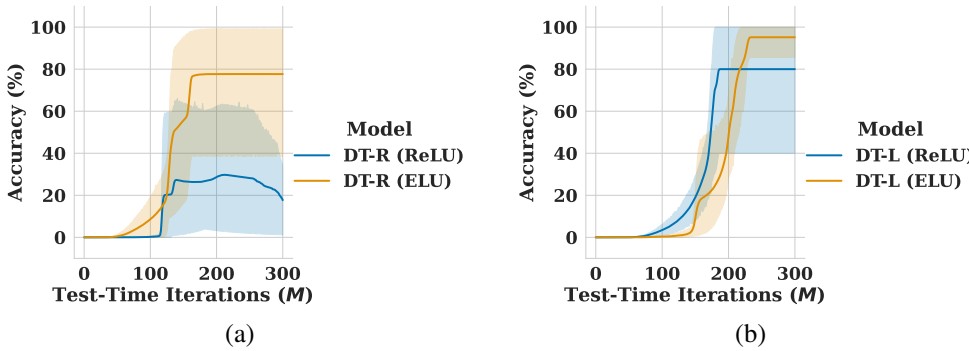

Figure D5: Mean solution accuracy at different $M$ for (a) 5 instances of DT-R (with ReLU activations) and 5 instances of DT-R (with ELU activations), and for (b) 5 instances of DT-L (with ReLU activations) and 5 instances of DT-L (with ELU activations). All models have width $w = 32$. Extrapolation performance tested on 512-bit prefix sum problem instances. Shaded areas show 95% confidence intervals.

**Bias in the Final Layer.**    DT-R has no bias terms throughout the entire model. We compare this to a version which has a bias term in *only* the final layer of the model.

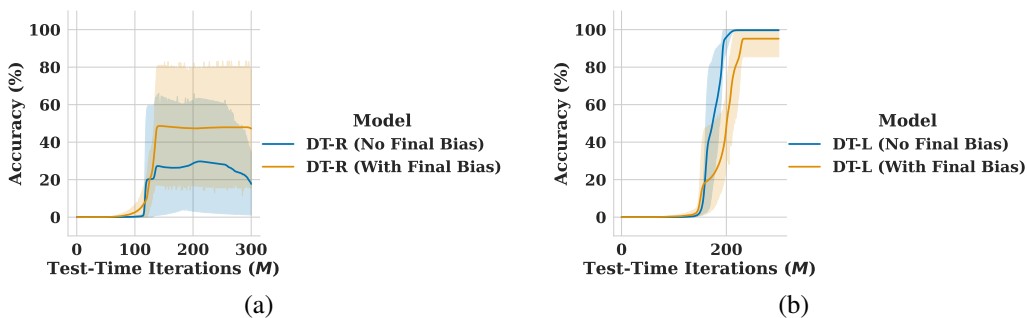

Figure D6: Mean solution accuracy at different $M$ for (a) 5 instances of DT-R (without final bias term) and 5 instances of DT-R (with final bias term), and for (b) 5 instances of DT-L (without final bias term) and 5 instances of DT-L (with final bias term). All models have width $w = 32$. Extrapolation performance tested on 512-bit prefix sum problem instances. Shaded areas show 95% confidence intervals.

**Batch Normalization.**    DT-R has no batch normalization layers in the model. We perform a study to measure the impact of adding batch normalization to DT-R (Figure D7(a)), as well as remove it from DT-L (Figure D7(b)).

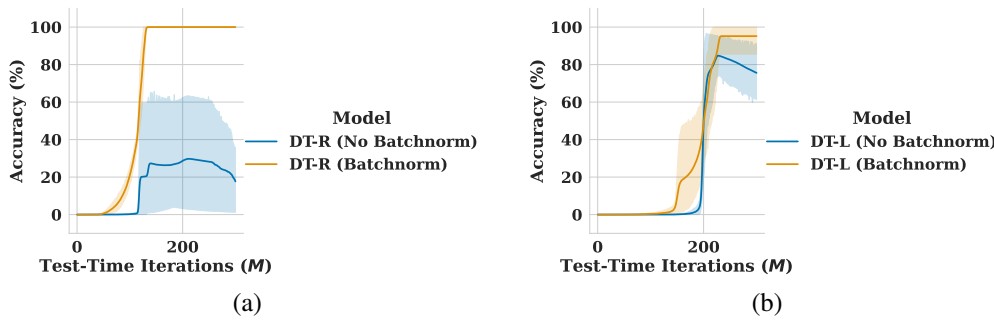

(a)                                                            (b)

Figure D7: Mean solution accuracy at different $M$ for (a) 5 instances of DT-R (without batch normalization) and 5 instances of DT-R (with batch normalization), and for (b) 5 instances of DT-L (without batch normalization) and 5 instances of DT-L (with batch normalization). All models have width $w = 32$. Extrapolation performance tested on 512-bit prefix sum problem instances. Shaded areas show 95% confidence intervals.

# E   Running Times and Peak Memory Usage

After all experiments had been completed, a modified version of the spectral normalization code was created which caches the normalized weights. This modification gives improved training times.

Table E1 provides the training times of DT-R and DT-L *before* this modification (the version used in the experiments). Table E2 provides the training times of DT-L *after* this modification.

Table E1: Representative training times and peak memory usage for different models. Training time is given as `hours:minutes`.

| Model | Problem | Batch Size | Epochs | Hardware | Training Time | Memory Usage |
|---|---|---|---|---|---|---|
| DT-R $w = 32$ | Prefix Sums | 500 | 150 | RTX8000 | 00:10 | 1.25 GB |
| DT-L $w = 32$ | Prefix Sums | 500 | 150 | RTX8000 | 00:30 | 1.34 GB |
| DT-R $w = 32$ | Mazes | 50 | 50 | RTX8000 | 03:30 | 4.30 GB |
| DT-L $w = 32$ | Mazes | 50 | 50 | RTX8000 | 04:02 | 4.41 GB |
| DT-R $w = 16$ | Chess Puzzles | 300 | 120 | A100 | 02:55 | 10.87 GB |
| DT-L $w = 16$ | Chess Puzzles | 300 | 120 | RTX8000 | 07:36 | 10.67 GB |

Table E2: Representative training times and peak memory usage for different models (using spectral normalized weight caching). Training time is given as `hours:minutes`.

| Model | Problem | Batch Size | Epochs | Hardware | Training Time | Memory Usage |
|---|---|---|---|---|---|---|
| DT-L $w = 32$ | Prefix Sums | 500 | 150 | RTX8000 | 00:19 | 1.30 GB |
| DT-L $w = 32$ | Mazes | 50 | 50 | RTX8000 | 03:15 | 4.42 GB |
| DT-L $w = 16$ | Chess Puzzles | 300 | 120 | RTX8000 | 04:47 | 10.70 GB |

