# OpenReview forum: "Rethinking Deep Thinking: Stable Learning of Algorithms using Lipschitz Constraints"
_NeurIPS.cc/2024/Conference — NeurIPS 2024 poster_

### Official Review · Reviewer_vZtn · 2024-06-13

**Soundness:** 3
**Presentation:** 3
**Contribution:** 2
**Rating:** 6
**Confidence:** 3

**Summary:**

To solve the stability of Deep Thinking models, this paper proposes to constrain activation functions to be Lipshitz-1 functions. The original DT and DT-R models have training stability problem, basically because of scale explosion or vanishing. The authors revealed the stability problem, attribute the problem to Lipschitz constants, proposed ways to ensure Lipschitz smoothness, and show the effectiveness of their approach through a few examples used in the original DT paper, as well as include the traveling salesman problem.

**Strengths:**

* This paper is clearly written and well motivated.
* The storyline is very reasonable: identify problems => propose ways to solve the problem => show the approach actually works
* This approach is mathematically grounded.
* Experiments are thorough, by running many random seeds and report error bars.

**Weaknesses:**

* The idea is quite straight-forward (may not be a bad thing, but make technical contributions smaller)
* In the TSP problems, DT-L's results seem worse than NN Tours and BNN Tours. At least some explanation is warranted.
* I'm not fully convinced by the significance of this paper. The examples shown in the paper are quite toy. Are there more examples you expect DT-L would work?
* I'd appreciate more visualizations that can intuitively show the benefits of DT-L over DT/DT-R? Maybe some Figures like in the original DT paper.
* The title is not very informative. Might be better to mention Lipschitz smoothness in the title.

**Questions:**

* In Line 141-142, I don't quite get this comment "Although any Lipschitz constant less than 1 would guarantee convergence, the nature of the problem solving mechanism we seek to learn intuitively means that we do not want fast convergence." Why don't we want faster convergence.
* In Figure 6 left, it looks like DT-L is worse than DT-R? Why is that? More stability leads to worse performance?
* What about DT and DT-R for TSP?

**Limitations:**

The authors cleraly addresses limitations.

---

> ### Author Rebuttal · Authors · 2024-08-06
>
> We thank the reviewer for taking the time to read the paper and for raising some important points in regards to our submission that we agree should be addressed.
>
> ### Response to Weaknesses
>
> > *"The idea is quite straight-forward (may not be a bad thing, but make
> > technical contributions smaller)"*
>
> We believe that imposing a sub-Lipschitz-1 constraint provides guarantees about the behaviour of DT-L architecture that previous models lacked (e.g. we are guaranteed that the recurrence will converge to a unique fixed-point).  We strongly believe this opens up avenues to future improves that would not be possible otherwise.  Finally, if the reviewer was seeking technical innovation we would point to our TSP model that involve a number of novel tricks that we are quite happy about (this is incidental to where we believe the main contribution of the paper lies so we have not emphasized these).
>
>
> > *"In the TSP problems, DT-L's results seem worse than NN Tours and BNN Tours.
> > At least some explanation is warranted."*
>
> We have responded to this comment in the response to all authors.  We fully agree that further explanation is warranted.  We had understated the difficulty of solving TSP using a convolutional algorithm that only gets to see a small part of the distance matrix.
>
> > *"I'm not fully convinced by the significance of this paper. The examples
> > shown in the paper are quite toy. Are there more examples you expect DT-L
> > would work?"*
>
> In this paper we deliberately chose to make DT-L as close to DT-R as possible to allow easy comparison and to focus attention on the benefits of imposing the sub-Lipschitz-1 constraint.  However, this constraint (with its theoretical underpinnings) allows a whole host of modifications to deep thinking type architectures to be developed that will train robustly.  We attempted to demonstrate this by learning an algorithm to tackle random TSP instances.  We believe (and have strong evidence based on work conducted after submitting this paper) that there is considerable potential to tackle many tasks considered 'real' world.  Given the page constraints of the paper, and a desire not to make this paper any more complicated, we feel going into this is outside the scope of this work.
>
> > *"I'd appreciate more visualizations that can intuitively show the benefits of
> > DT-L over DT/DT-R? Maybe some Figures like in the original DT paper."*
>
> The main benefits of DT-L are the stability it provides, especially in the case
> of smaller models. The focus is not on the algorithms it may learn or any
> attempt at interpreting them. While these are interesting visualizations to
> produce, this is not the topic of the paper and we believe including such
> visualizations may detract from the main contribution.
>
> > *"The title is not very informative. Might be better to mention Lipschitz
> > smoothness in the title."*
>
> We agree with the reviewer that the title is not very informative, and are
> pleased to incorporate the suggestion given by amending the title to
> **Rethinking Deep Thinking: Stable Learning of Algorithms using Lipschitz
> Constraints**.
>
> ### Answers to Questions
>
> > *"In Line 141-142, I don't quite get this comment "Although any Lipschitz
> > constant less than 1 would guarantee convergence, the nature of the problem
> > solving mechanism we seek to learn intuitively means that we do not want fast
> > convergence." Why don't we want faster convergence."*
>
> This is a good question, and something we will clarify in the final version of
> the paper.
>
> Our reasoning behind promoting slower convergence is to obviate the learning of trivial solutions to the given problem. As the model is trained on simple problems, we need to avoid it finding a simple solution (e.g. a shortcut, or memorisation) that does not generalise to larger problems. We re-emphasize that when we refer to convergence here, we are talking about the run-time convergence of the learned algorithm, not the training convergence (which we clearly do want to be fast and stable - something that our proposed solution also offers).
>
> Recall that we seek a solution using convolutions that only have access in a single iteration to information in a limited field of view.  To obtain good solutions requires information across the whole problem instance.  This is why to solve larger instances we need to run for more iterations.  Thus, allowing the solution to converge slowly (which we achieve by making the Lipschitz constant close to 1) prevents the network finding a sub-optimal solution requiring only local information.
>
> > *"In Figure 6 left, it looks like DT-L is worse than DT-R? Why is that? More
> > stability leads to worse performance?"*
>
> It is intriguing why DT-L performs slightly worse than DT-R on the chess problem and whether we pay a price for stability. It may well be that adding an additional constraint leads to a slight decrease in performance, however, these networks are sufficiently complex that we are reluctant to come to any firm conclusions without a lot of evidence. As an aside, we can easily modify DT-L networks to get better performance than the DT-R network. We have not put in these results as it is not a fair comparison (the network architectures are slightly different). To make a fair comparison we should also optimally modify DT-R, but DT-R is far harder to modify as many modifications lead to networks that don't train at all.
>
> > *"What about DT and DT-R for TSP?"*
>
> TSP is such a complex problem that getting the DT-R model to learn (rather than consistently diverge) is a huge challenge.  Of course, there is a huge hyper-parameter space to explore where it is possible that DT-R might work and we will endeavour in the appendix of the final version to add results for DT-R if we are ever able to train it (experiments underway at the moment).  Realistically DT is highly unlikely to ever solve this problem (DT-R dominates DT on almost all problems).

---

> > ### Comment · Reviewer_vZtn · 2024-08-11
> >
> > Thanks for addressing my concerns, I'll raise my score to 6.

---

### Official Review · Reviewer_9kC6 · 2024-07-10

**Soundness:** 4
**Presentation:** 4
**Contribution:** 3
**Rating:** 8
**Confidence:** 4

**Summary:**

This paper identifies and rectifies an issue with a particular type of iterative neural network called Deep Thinking Networks. The problem arises in exploding latent representations and unstable training routines. The authors of this work propose an update to the architecture where they add Lipschitz constraints to the model. They show three major benefits: (I) The models train more stably/predictably; (II) the inference-time behavior is better as the latent representations converge with iterations of the recurrent model; and (III) this new approach can learn how to solve NP-Hard problems where the old methods fail.

**Strengths:**

1. This paper is original to my knowledge. I am aware of much of the work on Deep Thinking Networks and the issues raised and the solutions proposed in this work are novel.
1. The quality of the work is high. For the most part the experiments are done well and cover many natural questions that would arise from reading the abstract/intro.
1. The clarity is good. I think the writing is clear and the results are compelling.
1. The results are significant for those interested in easy-to-hard generalization. These Deep Thinking Networks have strong extrapolation of toy problems and with the proposed updates to the methods they show strong performance even for TSP solving.

**Weaknesses:**

1. Clarity: A couple things could be more clear.
  i. I think IPT stands for Incremental Progress Training, but I don't see the acronym defined anywhere.
  ii. Table 1 the units are unclear. I gather there are tour lengths, but that isn't stated in the table or the caption.
  iii. The violin plot in Figure 2 is hard to parse (no harder than any other violin plot). This type of graphic does look nice, but offers little quantitative context. For example, there is no indication of the units/scale of the width of each violin. This is not the right type of plot for a conference paper.

**Questions:**

1. Can the authors make the clarifications needed to address my first two points in the Weaknesses section?
1. Have the authors looked at transformer architectures at all? I'm not asking for results to be added to the paper, but I'm curious about how these techniques, which are independent from the parameterization of any given layer in some ways, might apply to modern large model architectures.

**Limitations:**

Yes, the limitations are adequately addressed.

---

> ### Author Rebuttal · Authors · 2024-08-06
>
> We would like to thank the reviewer for their support and helpful suggestions, and are glad they are as excited about this direction of research as we are.
>
> ### Response to Weaknesses
>
> > *"Clarity: A couple things could be more clear.
> > i. I think IPT stands for Incremental Progress Training, but I don't see the
> > acronym defined anywhere.
> > ii. Table 1 the units are unclear. I gather there are tour lengths, but that
> > isn't stated in the table or the caption.
> > iii. The violin plot in Figure 2 is hard to parse (no harder than any other
> > violin plot). This type of graphic does look nice, but offers little
> > quantitative context. For example, there is no indication of the units/scale
> > of the width of each violin. This is not the right type of plot for a
> > conference paper."*
>
> We have updated the paper to appropriately define 'incremental progress
> training' (IPT) and have updated the caption of Table 1 to specify that the DT-L
> column consists of tour lengths. These were both oversights on our behalf, and
> we thank the reviewer for pointing these out.
>
> The violin plot (Figure 2) will be updated with a different graphic which better
> shows the distribution of singular values in reshaped kernel weights, providing
> clearer quantitative insight. We have provided an alternative graphic (Figure
> R1) in the PDF uploaded to the overall rebuttal, and trust that you find this more appropriate?
>
> ### Answers to Questions
>
> > *"Can the authors make the clarifications needed to address my first two
> > points in the Weaknesses section?"*
>
> See above.
>
> > *"Have the authors looked at transformer architectures at all? I'm not asking
> > for results to be added to the paper, but I'm curious about how these
> > techniques, which are independent from the parameterization of any given layer
> > in some ways, might apply to modern large model architectures."*
>
> Yes, we have considered this. We believe it is a challenging problem that is
> yet to be solved and have started to give it some thought.

---

> > ### Comment · Reviewer_9kC6 · 2024-08-12
> > **Response to authors**
> >
> > Thank you for addressing my points. I'll maintain my score.

---

### Official Review · Reviewer_x7iE · 2024-07-11

**Soundness:** 3
**Presentation:** 3
**Contribution:** 1
**Rating:** 6
**Confidence:** 4

**Summary:**

The paper addresses the positive feedback issue in the so called Deep Thinking networks, where the inference computation may involve more recurrent computations than encountered in training.  The proposed solution is to normalise the state vector that undergoes the recurrence, i.e. make the mapping contractive, i.e. ensure negative (but just) feedback.

**Strengths:**

The paper is well written and clear to follow, the proposed method is pretty straight forward and effective.

**Weaknesses:**

As far as I can tell, it is pretty straight forward control theory stuff for addressing positive feedback.  Nothing wrong with the proposed solution, but I would assume this is such a fundamentally well known issue in any recurrent/feedback system that we can leave this to be addressed by the designer at implementation time with any choice of normalisation.  It is somewhat disappointing that with the proposed method there is still the need for batch normalisation.

**Questions:**

Does batch normalisation alone not do a good job of stabilising the feedback?

**Limitations:**

If I understand this correctly, the proposed normalisation creates vanishing gradient problem, but authors seem to be aware of this and address it by keeping the spectral norm close to 1.

---

> ### Author Rebuttal · Authors · 2024-08-06
>
> We thank the reviewer for their comments.
>
> ### Response to Weaknesses
>
> > *"As far as I can tell, it is pretty straight forward control theory stuff for
> > addressing positive feedback. Nothing wrong with the proposed solution, but I
> > would assume this is such a fundamentally well known issue in any
> > recurrent/feedback system that we can leave this to be addressed by the
> > designer at implementation time with any choice of normalisation. It is
> > somewhat disappointing that with the proposed method there is still the need
> > for batch normalisation."*
>
> We agree that it is well-known that any learned recurrent system can explode or vanish.  The most prominent method for solving this is to have memory cells that are mainly conserved as exemplified by LSTMs and their variants.  This mechanism does not fit the DT architecture.  Imposing a sub-Lipschitz-1 constraint to ensure that the recurrence reaches a unique solution is to the best of our knowledge the first time this has been done in the context of recurrent networks (and certainly in the case of deep thinking networks).  It is, in our experience, somewhat non-trivial to ensure this constraint holds in a network with learnable parameters.  As shown by the previous DT papers, ensuring Lipschitz-1 behavior is not necessary for the models to work, although it makes the model much more fragile (i.e. it often fails to learn).  The contribution of this paper is showing that imposing a Lipschitz constraint cures this problem.  There are a number of unexpected consequences of doing this.  For example, we show we are able to reliably solve the problems with many fewer parameters (often by orders of magnitude).  The added stability makes it far easier to work with these models, to the extent that we were able to get these models to find an algorithm for solving TSP.  Thus, in our view, the paper shows some unexpected consequences of controlling the norm which was not so well known that it had been implemented in previous models.  As discussed below, batch normalization is not used in the recurrent section of the network (network $\mathcal{G}$) and, in fact, it causes problem when it is added (perhaps illustrating that controlling the norm is less trivial than it may appear).
>
> ### Answers to Questions
>
> > *"Does batch normalization alone not do a good job of stabilising the
> > feedback?"*
>
> As stated in the paper, the batch norm is not used in the recurrent
> part of the model. Instead, batch normalization is only used outside of the
> recurrent part. If the reviewer believes this is unclear in the current version,
> we can update it to be clearer about where in particular batch normalization is
> used.
>
> Our reasons for not using batch normalization in recurrence follows from
> existing literature, particularly 'Residual Connections Encourage Iterative
> Inference' (Jastrzebski *et al.*, 2018, DOI: 10.48550/arXiv.1710.04773) where
> _unsharing_ batch normalization
> statistics was necessary to avoid activations exploding. In an architecture
> where the maximum number of iterations is unknown (and potentially large in the
> cases of large prefix sum and maze problems), it appears infeasible to unshare
> batch normalization statistics for every iteration. Nonetheless, we look forward
> to future research that may resolve this issue.  It is also worth mentioning that batch norm does not ensure that the model is constrained to be sub-Lipschitz-1, which is another reason we have not used it.

---

> > ### Comment · Reviewer_x7iE · 2024-08-11
> > **Response to authors**
> >
> > Thanks for answering my questions, and the clarification of where batch norm fits in the proposed solution.  And as a result I grant that this work should be judged more on the way it tackles the straight forward/obvious approach (so that the model is stable and still capable of learning) as opposed to the fact that it uses the obvious approach...  and so I am raising my score to 6.

---

### Official Review · Reviewer_ipKY · 2024-07-13

**Soundness:** 3
**Presentation:** 3
**Contribution:** 3
**Rating:** 6
**Confidence:** 4

**Summary:**

The paper introduces Deep Thinking with Lipschitz Constraints (DT-L), an improved version of the Deep Thinking (DT) networks, designed to enhance the stability and performance of iterative algorithm learning models. The authors address the instability issues inherent in DT networks by analyzing intermediate representation growth and applying Lipschitz constraints. The DT-L model guarantees convergence to a unique solution and demonstrates robustness in learning algorithms that extrapolate to more complex problems. The paper furthermore benchmarks DT-L on the Traveling Salesperson Problem (TSP) as well other than the datasets used in the Deep Thinking models. It compares its performance against existing DT models.

**Strengths:**

- Introducing Lipschitz constraints into the DT framework enhances the models' reasoning capabilities. This approach addresses instability issues in training and inference, offering theoretical guarantees for convergence.
- DT-L demonstrates the ability to scale to larger problems effectively, maintaining stability and performance, which is crucial for real-world applications.
- The comprehensive evaluation on various problem classes, including prefix sums, mazes, chess puzzles, and TSP, highlights the robustness and versatility of the DT-L model.
- The paper provides a thorough analysis of the issues with DT networks and clearly explains how the proposed modifications address these problems.

**Weaknesses:**

- The modifications and theoretical underpinnings of the DT-L model, such as the Lipschitz constraints and orthogonal transformations, add complexity to the model, which might hinder its adoption and understanding by a broader audience.
- While the DT-L model shows improvement, its performance on the TSP is not impressive, indicating room for further optimization and refinement.

**Questions:**

- How does the introduction of Lipschitz constraints impact the computational complexity and training time of the DT-L model compared to traditional DT models?
- Can the proposed DT-L model be extended to other types of iterative algorithms beyond the ones tested in this paper? If so, what modifications would be necessary?
- Can this applied for transformer architectures like looped transformers?
- Can the insights gained from this work be applied to improve the interpretability of the learned algorithms, making the decision-making process of the DT-L model more transparent?

---

> ### Author Rebuttal · Authors · 2024-08-06
>
> We thank the reviewer for their positive and constructive comments.
>
> ### Response to Weaknesses
>
> > *"The modifications and theoretical underpinnings of the DT-L model, such as
> > the Lipschitz constraints and orthogonal transformations, add complexity to
> > the model, which might hinder its adoption and understanding by a broader
> > audience."*
>
> The main contribution of the paper in our view is that by imposing a Lipschitz constraint the recurrence is guaranteed to converge leading to much more stable training.  We believe that this is conceptually simple enough and provides sufficient benefits that it will be readily adopted by the broader community. However, as you have acknowledged there is complexity to this. The orthogonal transformation is only used for the TSP-solving variant of DT-L.  The complexity of solving TSP and ensuring that the tour constraint is met is sufficiently difficult that whenever tackling TSP some technical complexity is inevitable.  For solving the other problems the orthogonal transformation is not used and the network design was chosen to be as close to that of DT-R that we could make it.
>
> > *"While the DT-L model shows improvement, its performance on the TSP is not
> > impressive, indicating room for further optimization and refinement."*
>
> We have provided a response to this weakness as part of our response to all reviewers.  We would emphasize that finding an algorithm capable of solving random instances of general TSP is a surprisingly challenging problem.
>
> ### Answers to Questions
>
> > *"How does the introduction of Lipschitz constraints impact the computational
> > complexity and training time of the DT-L model compared to traditional DT
> > models?"*
>
> The implementation we have used for spectral normalization computes the spectral
> norm (from cached power iteration values) and divides the weights by this value
> for every weight access. This is computationally expensive, but is a simple
> addition to the model with PyTorch's parametrization features. Since the
> submission of this paper, a modification to our implementation of spectral
> normalization allows caching of these weights while maintaining gradient
> information. This has allowed a significant improvement to training speed, which
> we intend to show by adding an entry of improved training time to Table E1 for
> each DT-L model.
>
> We would also point out that due to the added stability we get by imposing the Lipschitz constraint we are able to tackle the test problems with significantly smaller networks than those used in the DT-R paper.  This provides a considerable improvement in speed making training these networks a lot more efficient than the larger DT-R models.  If we attempt to use smaller versions of the DT-R model we found that the model almost always failed to train properly.
>
> Finally, because of the increased stability we generally only have to train our model once to obtain a working solution. In contrast there are instances where we had to train DT-R over 20 times in order to find a solution that worked.
>
> > *"Can the proposed DT-L model be extended to other types of iterative
> > algorithms beyond the ones tested in this paper? If so, what modifications
> > would be necessary?"*
>
> We believe extensions to other types of problems and implementations are
> technically possible, but they go beyond the remit of this paper. In particular we see
> DT-L as being suitable for problems which naturally lend themselves to being
> solved by repeated convolutions, but extending the architecture to other
> problems is something we are considering and working towards.
>
> > *"Can this applied for transformer architectures like looped transformers?"*
>
> There are challenges to overcome in adapting transformer architectures to this
> application. It would be fantastic to see if one can create an iterative
> transformer with these constraints in the context given by the paper. We have
> been giving it some thought, but are not at the point where we have results. A
> transformer-based version of DT-L is out-of-scope, but this paper provides a
> critical stepping-stone towards this goal.
>
> > *"Can the insights gained from this work be applied to improve the
> > interpretability of the learned algorithms, making the decision-making process
> > of the DT-L model more transparent?"*
>
> This is a really good question, and something we're looking into.
> We're currently working on this since we believe guaranteeing uniqueness to a
> solution (from being a contractive mapping) should improve interpretability, but
> this is not work for this paper.

---

### Author Rebuttal · Authors · 2024-08-06

We would like to thank all reviewers for their careful reading of the paper and their insightful comments. We are pleased that overall the reviewers found the paper clear, but we will integrate the helpful suggestions that have been made - thanks!

We have responded to the reviewers individual comments separately, but there were a two general comments made by multiple reviewers that we address here.

**A number of reviewers commented on the performance of TSP.**  A previous criticism of deep thinking papers, one we don't share, is that the problems were cherry picked for the architecture.  We chose TSP as a problem we knew to be notoriously hard, but fitted the requirement of a problem that we could scale up to a large size to create a harder problem.  Any implementation of a TSP solver is challenging because of the tour constraint.  We wanted to demonstrate that DT-L was sufficiently robust that it could find an algorithm for solving TSP.  To increase the challenge we chose non-symmetric (and obviously non-Euclidean) TSP, and additionally we do not provide the optimal solution during training, utilizing only a loss that penalizes tour length.  In this case many of the classic heuristics fail.  For example, k-OPT which is widely used in hill-climbing type solvers are inefficient as they typically involve reversing part of the tour which completely changes the path length.  Other common heuristics used for Euclidean TSP (such as locality of cities) also fail for non-Euclidean problem instances.  Thus, nearest neighbor algorithms actually provide a stiff baseline for this problem.  Note, however, that DT-L has only a local view of the distance matrix as it is using convolutions.  Nearest neighbor in contrast is a constructive algorithm that requires global information of the distance matrix (it has to find the closest city that has not already been included in the tour).  Thus, although the results we obtain on TSP may appear disappointing at first view we believe that the fact that they are considerably better than random tours shows DT-L's ability to learn non-trivial algorithms.  Finally, we should mention that for comparison with previous DT models we have deliberately stuck to the same architecture as DT-R as much as possible (although, admitted we needed to make some modification to get TSP to learn at all).  There are additional modification such as increasing the number of layers in the recurrent section of the network that leads to improved performance on TSP, but we have not included these in the current paper as we believe presenting too many modifications to the network would obscure the main contribution of adding the Lipschitz constraint.  We do agree with the reviewers that a comment on the TSP results would be useful and we will modify our text to include this.

**The reviewers mentioned extending to transformers and other tasks.** We are looking forward to exploring this direction in future work, but this is not the focus of this paper. The ideas presented here are however critical to actually extending deep thinking style _recurrent_ networks to other tasks. Our work also shows what design constraints of the recurrent architecture are required to ensure convergence, leading us to be able to propose new architectures in a principled way in the future. We have expanded more in the responses to individual reviewers.

---

### Decision · Program_Chairs · 2024-09-25

**Decision:**

Accept (poster)

**Comment:**

**Summary:** This paper presents an enhancement to Deep Thinking networks by introducing Lipschitz constraints, leading to the development of the DT-L model. The key contribution is the imposition of a sub-Lipschitz-1 constraint, which guarantees convergence to a unique solution and addresses stability issues that have plagued previous DT models. The authors demonstrate the robustness of DT-L across several tasks, including the Traveling Salesperson Problem, and provide theoretical and empirical support for their claims.

**Strengths:**
- The paper tackles a significant issue in DT networks—training instability—by proposing a theoretically grounded solution in the form of Lipschitz constraints. This addition provides stability and guarantees convergence, addressing a fundamental problem in iterative algorithm learning.
- The authors conduct thorough experiments across various tasks, including TSP, prefix sums, and maze-solving, demonstrating the versatility and robustness of DT-L.
- The paper is well-written, with a clear narrative that connects the identification of the problem to the proposed solution and its validation through experiments.

**Weaknesses:**
- Technical Contribution: Some reviewers noted that while the application of Lipschitz constraints is effective, it may be seen as a straightforward application of well-known control theory concepts. This perception could limit the perceived novelty of the contribution.
- Performance on TSP: The results on the TSP benchmark are not as impressive as expected, with DT-L performing worse than some baseline methods. The authors provided an explanation, but the performance gap remains a concern.
- Evaluation and Generalization: Although the paper includes several experiments, there are concerns about the generalizability of the results. The tasks tested are somewhat toy-like, and it is unclear how DT-L would perform on more complex real-world tasks.

**Unresolved Issues:**
- Comparison with Baselines: The performance of DT-L on TSP was below expectations, and the authors’ explanation, while reasonable, does not fully resolve the concern. A more thorough comparison with DT and DT-R on TSP would strengthen the evaluation.
- Broader Impact: The paper’s focus on stability and convergence is valuable, but the broader impact on more diverse and real-world tasks is not fully explored. Further exploration in future work could address this limitation.

**Conclusion:** The paper provides a significant contribution to the field of iterative algorithm learning by enhancing the stability and convergence of DT networks through Lipschitz constraints. Despite some concerns about the novelty and generalization of the approach, the theoretical guarantees and robust empirical results make this a valuable addition to the literature. The paper is recommended for acceptance, with the expectation that future work will address the unresolved issues and explore broader applications of the proposed method.